# The Clinical Effect of Steroids for Hearing Preservation in Cochlear Implantation: Conclusions Based on Three Cochlear Implant Systems and Two Administration Regimes

**DOI:** 10.3390/ph15101176

**Published:** 2022-09-22

**Authors:** Magdalena B. Skarżyńska, Aleksandra Kołodziejak, Elżbieta Gos, Adam Walkowiak, Artur Lorens, Andrzej Pastuszak, Łukasz Plichta, Piotr H. Skarżyński

**Affiliations:** 1Institute of Sensory Organs, Mokra 1, 05-830 Kajetany, Poland; 2Center of Hearing and Speech Medincus, Mokra 7, 05-830 Kajetany, Poland; 3World Hearing Center, Department of Teleaudiology of Hearing, Institute of Physiology and Pathology of Hearing, Mokra 17, 05-830 Kajetany, Poland; 4World Hearing Center, Department of Cochlear Implants, Institute of Physiology and Pathology of Hearing, Mokra 17, 05-830 Kajetany, Poland; 5World Hearing Center, Oto-Rhino-Laryngology Surgery Department, Institute of Physiology and Pathology of Hearing, 05-830 Warsaw, Poland; 6Heart Failure and Cardiac Rehabilitation Department, Faculty of Medicine, Medical University of Warsaw, 03-242 Warsaw, Poland

**Keywords:** cochlear implantation, steroid administration, partial deafness treatment, dexamethasone, prednisone

## Abstract

The main aim of this study was to assess the clinical effect of steroids (dexamethasone and prednisone) on hearing preservation in patients who underwent cochlear implantation with different cochlear implant systems (Oticon^®^, Advanced Bionics^®^, Med-El^®^). 147 adult patients met the inclusion criteria and were enrolled to the study and divided into three groups depending on the brand of cochlear implant they received and participated in all follow-up visits regularly. They were also randomly divided into three subgroups depending on the steroid administration regime: (1) intravenous dexamethasone (0.1 mg/kg body weight twice a day for three days); (2) combined intravenous and oral steroids (dexamethasone 0.1 mg/kg body weight twice a day plus prednisone 1 mg/kg weight once a day); and (3) no steroids (control group). The results were measured by pure tone audiometry (PTA) at three time points: (i) before implantation, (ii) at processor activation, and (iii) 12 months after activation. A hearing preservation (HP) figure was also calculated by comparing the preoperative results and the results after 12 months. Further measures collected were electrode impedance and hearing threshold in the non-operated ear. The highest HP measures (partial and complete) were obtained in the subgroups who were given steroids. Of the 102 patients given steroids, HP was partial or complete in 63 of them (62%). In comparison, partial or complete HP was achieved in only 15 patients out of 45 (33%) who were not given steroids. There were differences between the three cochlear implant groups, with the Med-El and Advanced Bionics groups performing better than the Oticon group (45% and 43% of the former two groups achieved partial or complete HP compared to 20% in the latter). Hearing thresholds in the non-operated ear were stable over 12 months. Generally, impedance was slightly lower in the 12 month follow-up in comparison with the activation period, with the exception of the Oticon group. (4) Conclusions: Pharmacological treatment with steroids in patients undergoing cochlear implantation helps to preserve residual hearing.

## 1. Introduction

According to the current clinical indications for cochlear implantation (CI), which have been progressively expanded, candidates for the procedure are those with partial deafness (PDT, partial deafness treatment), children with congenital profound hearing loss, single-sided deafness patients, and those over 65 years of age, as well as those with acquired bilateral sensory hearing loss [1,2,3]. Some years ago, it would be very challenging, or even impossible, to preserve residual hearing after CI surgery [4,5]. Recent advances have been on the design of the electrodes, atraumatic surgical techniques, monitoring of cochlear function during implantation, and use of corticoids and other anti-inflammatory drugs, all of which help preserve residual hearing [6,7].

Information from two of the world’s foremost agencies responsible for drug authorization—the Food and Drug Administration (FDA) and the European Medical Agency (EMA)—indicate that there is no drug or medical device approved for delivery direct to the inner ear. There are some options for delivering drugs intracochlearly (via the oval or round window through injection, catheter/micropump, stapes surgery, or cochlear implant), although these are largely restricted to animal studies and prototypes [8,9]. There are thus two possibilities for delivering drugs to the inner ear: systemic or local. The administration of glucocorticoids (especially those with high anti-inflammatory properties) to preserve residual hearing during CI surgery has been described, although the results are mixed [3,10]. Nevertheless, according to current knowledge, steroids appear to have an important role in reducing post implantation fibrosis and loss of hearing due to electrode insertion trauma [11].

### Aim of the Study and Its Endpoints 

The main aim of the study was to assess the clinical effect of steroids (dexamethasone and prednisone) on hearing preservation in patients who underwent implantation with three different CI systems (Oticon^®^, Advanced Bionics^®^, and Med-El^®^). The primary endpoint was the degree of hearing preservation calculated using the hearing preservation equation (Figure 7), which basically measures the change in average hearing threshold after a CI relative to the preoperative period. We wanted to investigate how average rates of preservation varied depending on the type of implant used and the pharmacological treatment employed. A second endpoint was the stability of hearing in both the operated and non-operated ears. A third endpoint was the mean impedance of the electrodes, again as a function of type of implant and treatment regime. Each endpoint was analysed in comparison with a control group, one for each main group, where steroids were not administered.

## 2. Results 

### 2.1. Statistical Analysis

Sex and age differences between the subgroups were assessed with the chi-square test and the Kruskal-Wallis test. A Wilcoxon test was used to compare preoperative hearing thresholds with those obtained 12 months after CI activation. Additionally, within each implant type group, comparisons of hearing thresholds obtained 12 months after CI activation were made between the intravenous, oral + intravenous, and control groups (Kruskal-Wallis test and Mann-Whitney *U*-test). A chi-square test was used to assess the relationship between treatment regime and hearing preservation. Statistical significance was set at *p* < 0.05. The analysis was performed using IBM SPSS Statistics, v. 24.

### 2.2. Hearing Thresholds in the Operated Ear

Hearing thresholds were averaged across all frequencies from 0.125 to 8 kHz. Table 1 shows hearing thresholds in the operated ear obtained preoperatively, at CI activation, and 12 months after activation.

The results of the analysis showed that hearing thresholds (between pre-op and 12 months after cochlear implant activation) changed significantly regardless of which group the patients belonged to. The average change was from 5 dB HL to 16.4 dB HL.

In the Advanced Bionics group, hearing thresholds obtained 12 months after CI activation were similar in both steroid groups. However, hearing thresholds in the control group were significantly worse than those in the *i.v.* group (*U* = 45.5; *p* = 0.012) or in the oral + *i.v.* group (*U* = 48.0; *p* = 0.013).

In the Oticon group, hearing thresholds obtained 12 months after CI activation were different in each of the steroid groups (worse in the *i.v.* group than in the oral + *i.v.* group; *U* = 82.0; *p* = 0.012). Hearing thresholds in the control group did not differ significantly from those obtained in the two steroid groups.

In the Med-El group, hearing thresholds obtained 12 months after CI activation were similar in both steroid groups. But hearing thresholds in the control group were significantly worse than in the *i.v.* group (*U* = 79.0; *p* = 0.013) or in the oral + *i.v.* group (*U* = 56.5; *p* = 0.002).

The above results relate to average hearing thresholds. Figure 1 shows thresholds for all 11 frequencies at each of the three time points and for each of the treatment regimes.

### 2.3. Hearing Preservation

Hearing outcomes in terms of degrees of hearing preservation are shown in Table 2.

A statistically significant relationship between treatment regime and hearing preservation was found in the Oticon group (*χ*^2^ = 16.12; *p* = 0.013) as well as in the Med-El group (*χ*^2^ = 19.90; *p* = 0.003), but not in the AB group (*χ*^2^ = 7.57; *p* = 0.271). In the Oticon group, most cases of no measurable hearing were found in the *i.v.* (81%) and control group (60%), while the widest range in hearing status was observed in the oral + *i.v.* group. In the Med-El group, most cases of partial or complete HP were found in the oral + *i.v.* group (52.4% and 33.3% respectively) and in the *i.v.* group (54.5 and 22.8%), while in the control group most patients had no measurable hearing. The results in the AB group were fairly similar to those in the Med-El group, but overall the relationship was not statistically significant.

### 2.4. Hearing Thresholds in the Non-Operated Ear

Hearing thresholds in the non-operated ear were stable over 12 months. Mean changes between pre- and post-op ranged from 0.1 dB HL (in the *i.v.* group implanted with Advanced Bionics) to 3.6 dB HL (in the control group implanted with Oticon). However, only in one of the subgroups—the control group implanted with Advanced Bionics —was the change between pre- and post-op hearing thresholds statistically significant (Wilcoxon test = 2.53; *p* = 0.001). However, in this case the mean deterioration was only 2.6 dB, i.e., less than the audiometer resolution. Mean hearing thresholds (averaged across all 11 frequencies) in the non-operated ear are shown in Figure 2.

### 2.5. Electrode Impedance 

Electrode impedance was averaged within each group and is shown in Figure 3. Generally, there were no statistical differences between three treatment regimes: the intravenous group, the oral+ intravenous group and the control group. In the Advanced Bionics groups in all three treatment regimes, mean impedance at activation was about 4–6 kΩ, and at 12 months post CI activation it slightly decreased. This also applies to the Med-El groups, however for this device, the values were generally higher, averaging about 5–7 kΩ at activation. A completely different pattern of impedance change was found in the Oticon group. Impedance values were much lower and they increased at each electrode 12 months after CI activation.

## 3. Discussion

According to the introduction and the data from Food and Drug Administration (FDA) and European Medical Agency (EMA), as of now there is no approved drug or medical device with constant relieve of drug for wide-using for delivering directly to the inner ear. There are some possibilities to deliver the drug locally intracochlearly via the oval window or round window by injections, catheter/micropumps, stapes surgery or cochlear implants, but this is not a widely used way of administering of drugs (animal studies, prototypes) [8,9,12,13,14,15,16,17]. As a result, there are two possibilities of delivering drugs to the inner ear [8,18]. The first is the systemic administration or off-label use of drugs locally, e.g., during the cochlear implantation [19]. From the anatomical and physiological there are a few challenges in delivery drugs as a systemic administration, such as: (1) limited local vascularization in the inner ear, (2) possible side effects of systemic administration of drugs (e.g., after delivery high doses of corticoids), (3) the blood-cochlear barrier (blood-labyrinth barrier (BLB)), [20,21,22]. Drugs in the systemic administration may be delivered orally or intravenously [23]. The intratympanic administration of drugs is also used in practice but in inner ear drug delivery may be limited due to the clearance through the Eustachian tube and the presence of a round and oval window [23]. Due to the lack of possible routes of delivery to the inner ear, there are limits in possible pharmacological treatment or supportive pharmacological treatment of many otologic diseases such as SSNHL (Sudden Sensorineural Hearing Loss) or partial deafness.

The preservation of residual hearing after CI surgery is an important issue for patients. The aim of this study was to investigate the optimum type of steroid, dose, and duration of treatment in comparison with the control group (without steroid administration). The benefit of extended preoperative systemic steroids on hearing after cochlear implantation surgery has already been shown [10], but whether dexamethasone administration should begin one day or five days prior to the surgery remains unclear.

Dexamethasone has high anti-inflammatory and immune-modulative properties [24,25,26,27]. It plays an important role in reducing the concentration of inflammatory cytokines and promoting hair cell survival, and, in a guinea pig model, reducing severe intracochlear fibrosis and ossification [28]. Of significance is the fact that this study is not able to extend patient observations up to 12 months of follow-up. Moreover, it also investigates whether a combination of steroids is effective. This is important, as it opens up the possibility of the patients themselves administering prednisone orally after discharge from hospital. According to the literature and a systematic review, dexamethasone and triamcinolone are two glucocorticoids with high potency. In this respect, dexamethasone is better than triamcinolone, since the latter is more rapidly eliminated from perilymph [9].

All patients who were enrolled in this study had severe or profound hearing loss according to the Skarzynski’s classification of partial deafness treatment (PDT). Nevertheless, the HP results, as shown in Table 2, were widely different. Two possible reasons for this are the type of cochlear implant and the age of the patients. The oldest patients were from the Oticon group, which also had the worst results in terms of hearing preservation. In the Oticon group, the median age was 61.2 y in the intravenous subgroup, 64.4 y in the intravenous + oral subgroup, and 67.7 y in the control group. In comparison, the median age in the Advanced Bionics group was 51.2, 61.4, and 61.6 years, respectively, and in the Med-El group the respective ages were 48.8, 56.2, and 49.4 years.

It is clear from Table 2 that there is a difference between Oticon and Advanced Bionics and Med-El results. In the Oticon no measurable HP, or minimal HP, was found in 36 patients (72%). In the Advanced Bionics and Med-El groups, however, the comparable figures were 14 patients (35%) and 19 patients (33.3%), respectively. Looking at figures for partial or complete HP, the Oticon group achieved this for only 14 patients (28%), whereas the Advanced Bionics group had 26 patients (65%) and the Med-El group had 38 patients (66.7%). Importantly, in each group the most successful HP (partial or complete) was generally achieved in the steroid groups rather than the control group. The exception was the Oticon group, where non-measurable hearing was found in 31 patients (62%) regardless of whether they received steroid therapy or not. In the Advanced Bionics group, the two steroid subgroups had partial or complete HP in 19 patients (47.5%), a figure better than in the control group where the comparable figure was seven patients (17.5%). In the Med-El group, the results were better than all the other groups: partial or complete HP was found in 35 patients (61.4%) from both steroid subgroups in comparison with the control group where the corresponding figure was just three patients (5.3%). In summary, from the 147 patients who were enrolled in this study, HP was partial or complete in 63 patients (42.9%) in both steroid groups, and this can be compared with just 15 patients (10.2%) in the non-steroid group. The main differences between brands of cochlear implant electrode arrays are the number of active electrodes, electrode array length, electrode array diameter and area of electrode contacts. These differences don’t lead to different hearing outcomes. However, the differences in the area of electrode contacts are associated with differences in electrode impedances. For Oticon implants, the larger area of an electrode contact (around 0,5 mm 2 comparing to 0,15 in others) leads to the smallest initial impedances.

Similar comparisons were made when the results of changes in the hearing threshold over two different periods were examined: preoperatively and in the 12-month follow-up period. The hearing thresholds in both steroid groups (intravenous and intravenous + oral) were better than in the control groups, regardless of the type of CI system (except Oticon, where different results were obtained in both steroid groups, although the group with the combination of steroids still gained better results in the 12-month follow-up period). Hearing thresholds in the non-operated ears were stable over 12 months.

The measurement of impedance is routinely done after the cochlear implantation. The results of impedance monitoring check if electrodes work correctly and additionally to monitor the implant functioning during the fitting of the speech processor. As a rule, impedance was slightly lower in the 12 month-follow up in comparison to the activation period, with the exception of the Oticon group of patients. However, the comparison between different brands of cochlear implants should be done with caution. The current method of testing allows the determination of only the overall electrode impedance with the different contribution of its components: access resistance and polarization impedance. It was demonstrated that access resistance increased slowly over time, whereas polarization impedance decreased after the commencement of electrical stimulation after the activation of the CI system. The increase of access resistance is caused by the formation of a layer of fibrous tissue around the electrode within the cochlear canal, The decrease of polarization impedance is due to the fact that electrical stimulation appears to disperse the surface layer of protein which is formed on the surface of the electrode in the early phase after implantation. We can speculate that the method of testing applied in AB and Med-El is more sensitive to the decrease of polarization impedance and the method of testing used in Oticon facilitates the monitoring of an increase in access resistance. However, our results indicate that there is no difference in overall impedances tested with the current method at activation and 12 months post activation between the steroid and control groups.

### The Strengths and Limitations 

This work adds to our knowledge of hearing preservation in cochlear implantation and supports the pharmacological use of steroids. The 12 months of observation strengthen the validity of the findings, as does the high number of patients enrolled (147 patients with partial deafness with hearing loss in the high frequency range). This study has compared different types of cochlear implants (three types of cochlear implants) and different steroid administration, which is pioneering. The discrepancy between the number of participants in each groups may be a limitation, but appropriate statistical methods were applied for unequal groups in order to gain reliable results.

## 4. Materials and Methods

This study involved 147 patients who were implanted with three different systems: Advanced Bionics (*n* = 40; Slim J electrode), Oticon (*n* = 50; EVO electrode), and Med-El (*n* = 57; Flex 28, Flex 24, and Flex 20 electrodes). Each of the three groups was subdivided into two subgroups with different strategies for steroid administration (intravenous and intravenous + oral), as well as a separate third control group. Characteristics of the patients in terms of sex, age, and operated ear are presented in Table 3.

In each of the three groups (Advanced Bionics, Oticon, and Med-el), it was checked whether there were sex and age differences in each subgroup. In the Advanced Bionics group there were no statistically significant differences in terms of sex (*χ*^2^ = 2.65; *p* = 0.266). The same was true for the Oticon group (*χ*^2^ = 0.71; *p* = 0.677) and the Med-el group (*χ*^2^ = 0.21; *p* = 0.899).

In the Advanced Bionics group, some differences in terms of age were apparent; the patients in the intravenous group were slightly younger than other patients, *χ*^2^ = 5.99; *p* = 0.050. There were no statistically significant differences in terms of age in the Oticon group, (*χ*^2^ = 2.03; *p* = 0.363) and the Med-el group (*χ*^2^ = 2.56; *p* = 0.271).

The criteria for inclusion were as follows: (1) age above 18 years and qualified for cochlear implantation; (2) severe or profound hearing loss according to Skarzynski’s classification of partial deafness treatment (PDT) such that the patient received either: (a) PDT electrical-acoustic stimulation (PDT-EAS) or (b) PDT electrical stimulation (PDT-ES) as illustrated in Figure 4. In terms of pure tone averages, patients had hearing sound levels in the range of 10–120 decibels (dB) at sound frequencies of 125–250 hertz (Hz); sound levels of 35–120 dB and frequencies of 500–1000 Hz; sound levels of 75–120 dB and frequencies of 2000–8000 Hz. Exclusion criteria were as follows: (1) age below 18 years; (2) comorbidities (e.g., severe hypertension, oncological disease, diabetes mellitus, where steroids may be contraindicated); (3) other pharmacological treatment which may interact with steroids (e.g., some immunosuppressant drugs). Ethics Committee approval was obtained (IFPS: KB/06/2016).

The scheme and route of administration of steroids in each group was different. In subgroup 1, dexamethasone (DEX) was administered intravenously (*i.v.*) at a dose of 0.1 mg per kg body weight 30 min before the insertion of the electrode to the inner ear and subsequently twice a day (every 12 h) for the next three days (Figure 5). The single and daily maximum dose of dexamethasone was not exceeded and was in accordance with the summary of product characteristics (SPC) of Dexaven^®^.

In subgroup 2, steroid therapy consisted of dexamethasone (DEX) administered intravenously (*i.v.*) combined with orally administered (*p.o*.) prednisone. The administration regime was as follows. Three days prior to cochlear implantation, prednisone at a dose of 1 mg per kg body weight (*p.o.*) was administered. Then, 30 min before insertion of the electrode into the inner ear, and for the next three days, dexamethasone was administered *i.v.* every 12 h (twice a day). For the next three days, prednisone was also administered *p.o.* (1 mg of prednisone per kg body weight). After this time, the dose was reduced, as shown in Figure 6. The single and daily maximum doses of dexamethasone were not exceeded and were in accordance with the SPC of Dexaven^®^; the same applied for prednisone (Encorton^®^).

### Measures

Our methods followed those described by Skarzynska et al., 2018 [6]. The primary endpoint (outcome) was the average hearing threshold across 11 frequencies (0.125–8 kHz) using both octaves and half-octaves according to the International Organization for Standardization (ISO 8253-1:2010). All measurements were conducted in the same soundproof cabin by an experienced technician, using the same diagnostic audiometer, the Madsen Itera II (GN Otometrics, Taastrup, Denmark) with calibrated earphones (TDH-39P) (Telephonics, NY, USA). A second outcome measure was hearing preservation (HP), which was calculated by comparing hearing thresholds in the one-year post-operative period with the preoperative hearing thresholds according to the formula in Figure 7 [31].

The HP levels were then divided into three categories: minimal hearing preservation (minimal HP) 0–25%; partial hearing preservation (partial HP) 26–75%; and complete hearing preservation (complete HP) > 75%.

## 5. Conclusions

Steroid administration in patients who undergo cochlear implantation is beneficial in preserving residual hearing. Of the 102 patients given steroids, HP was partial or complete in 63 of them (62%). In comparison, partial or complete HP was achieved only in 15 patients out of 45 (33%) who were not given steroids. The type of electrode also appears to play a part: the best results were obtained using the Med-El and Advanced Bionics devices, although here age may have been a confounding factor, and atraumatic insertion of the electrode is always important. The most beneficial regime for HP is intravenous dexamethasone at a dose of 0.1 mg per kg body weight, although in some cases a combination of prednisolone and dexamethasone may provide even better results.

## Figures and Tables

**Figure 1 pharmaceuticals-15-01176-f001:**
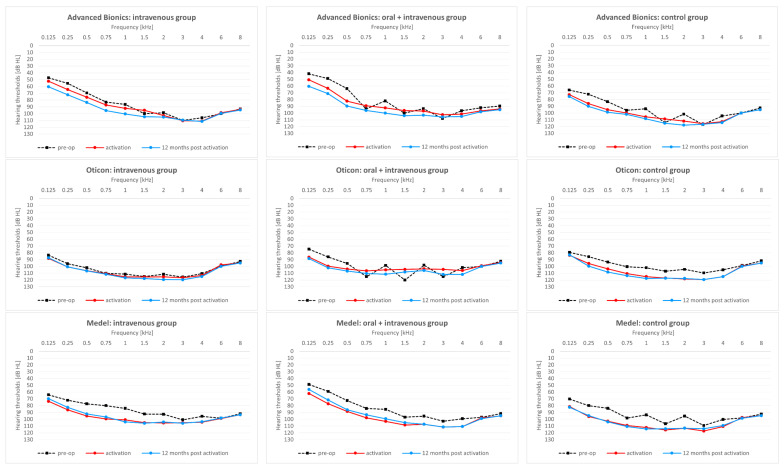
Hearing thresholds according to implant type and treatment regime before implantation, at CI activation, and 12 months after CI activation.

**Figure 2 pharmaceuticals-15-01176-f002:**
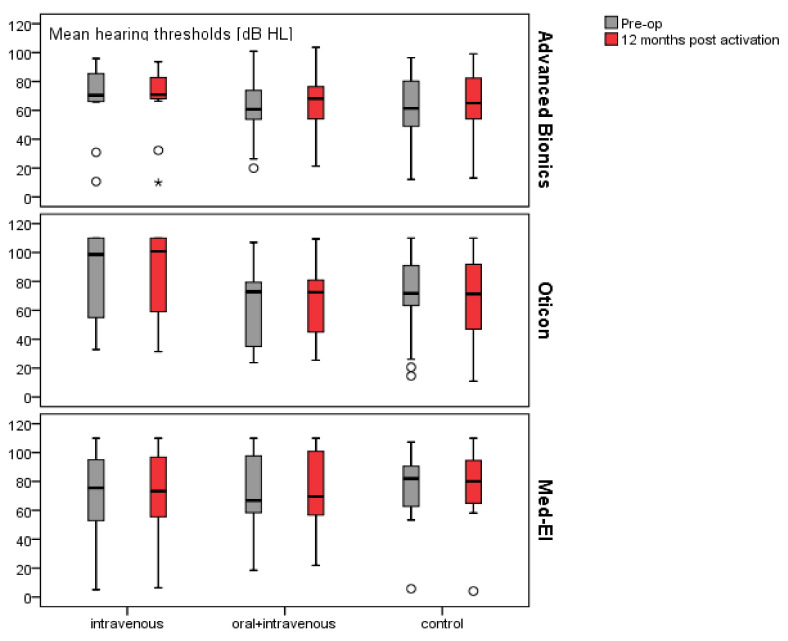
Mean hearing thresholds in the non-operated ear before implantation and 12 months after activation (asterisk “*” means outlier higher than three times than interquartile range (higher than 3 IQR and circle “o” means outlier between one, five and three times interquartile range (1,5 IQR–3 IQR).

**Figure 3 pharmaceuticals-15-01176-f003:**
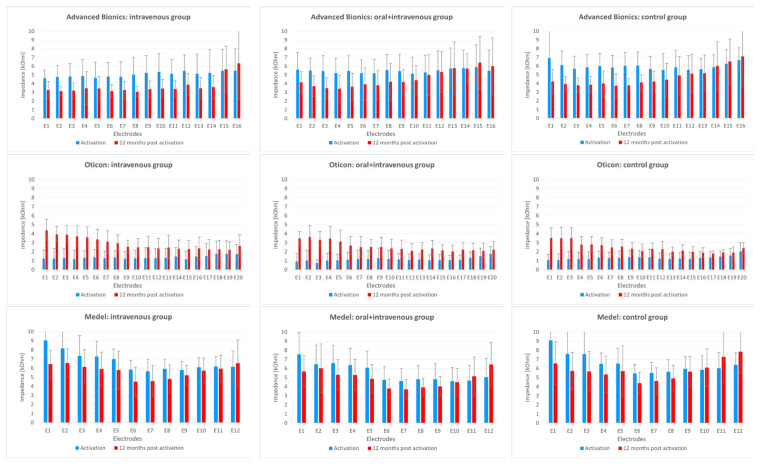
Mean impedance at all electrodes according to type of implant and treatment regime.

**Figure 4 pharmaceuticals-15-01176-f004:**
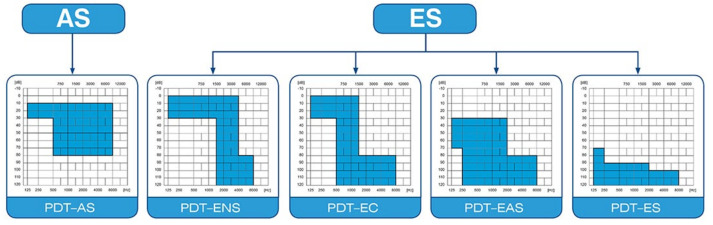
Partial deafness treatment groups for cochlear implantation. AS—acoustic stimulation; ENS—electro-natural stimulation; EC—electrical complement; EAS—electrical-acoustic stimulation; ES—electrical stimulation [3,29,30].

**Figure 5 pharmaceuticals-15-01176-f005:**
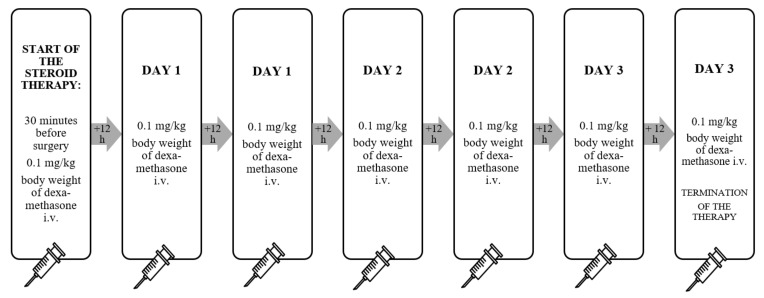
Scheme of administration of dexamethasone in subgroup 1 (*i.v.*—intravenous).

**Figure 6 pharmaceuticals-15-01176-f006:**
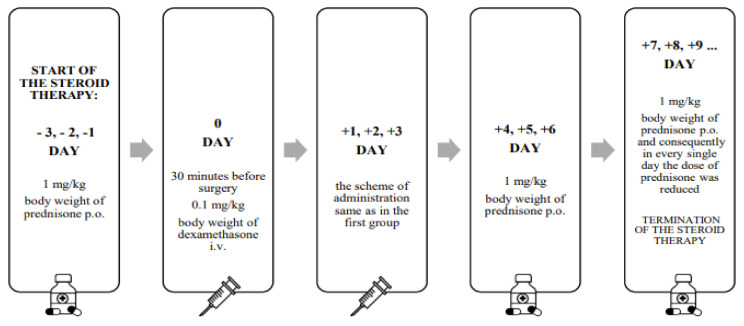
Drug regime for group 3, treated with combined oral (*p.o.*) and intravenous (*i.v.*) steroid therapy (prolonged therapy).

**Figure 7 pharmaceuticals-15-01176-f007:**
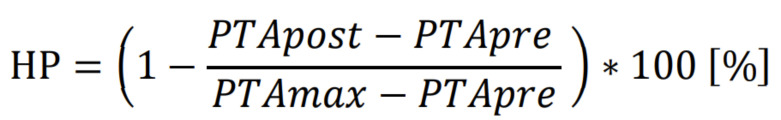
Hearing preservation (HP) equation. (PTApre is pure tone average measured preoperatively, PTApost is pure tone average measured postoperatively, and PTAmax is the maximum sound intensity generated by a standard audiometer, usually 120 dB).

**Table 1 pharmaceuticals-15-01176-t001:** Hearing thresholds (operated ear) in the patients at each time point.

	Group		Min	Max	M	SD	Me	Wilcoxon Test between Pre-Op and 12 Mo Post Act.	Mean Deterioration between Pre-Op and 12 Mo Post Act.
**ADVANCED BIONICS**	IV	Pre	67.50	97.73	83.30	10.20	81.25	2.93; 0.003	11.0 dB HL
		Activation	61.00	105.00	89.42	14.71	97.73		
		12 mo post activation	74.09	110.00	94.26	11.40	96.36		
	O+IV	Pre	55.00	100.00	77.11	11.55	72.27	3.18; 0.001	16.4 dB HL
		Activation	73.50	102.73	87.22	8.28	85.45		
		12 mo post activation	74.09	110.00	93.46	10.64	95.91		
	C	Pre	69.29	110.00	90.75	10.39	91.06	3.30; 0.001	12.3 dB HL
		Activation	83.00	110.00	100.31	7.68	99.55		
		12 mo post activation	87.73	110.00	103.01	7.89	105.45		
**OTICON**	IV	Pre	67.14	110.00	103.22	10.50	107.50	2.77; 0.006	5.0 dB HL
		Activation	86.50	110.00	106.88	5.95	110.00		
		12 mo post activation	87.73	110.00	108.23	5.13	110.00		
	O+IV	Pre	75.71	105.00	94.20	10.33	94.92	3.05; 0.002	10.5 dB HL
		Activation	86.00	110.00	101.17	7.64	102.27		
		12 mo post activation	91.36	110.00	104.74	6.42	106.82		
	C	Pre	83.18	110.00	97.18	9.59	98.00	3.30; 0.001	10.7 dB HL
		Activation	96.50	110.00	106.57	4.62	109.00		
		12 mo post activation	101.36	110.00	107.88	3.11	110.00		
**MED-EL**	IV	Pre	57.86	100.00	84.73	10.91	87.86	3.95; <0.001	11.2 dB HL
		Activation	55.00	109.09	97.07	12.77	102.05		
		12 mo post activation	59.55	110.00	95.97	13.53	97.0511		
	O+IV	Pre	67.50	101.36	83.32	10.06	81.3615	3.70; <0.001	10.6 dB HL
		Activation	72.14	110.00	95.98	8.76	96.36		
		12 mo post activation	82.73	110.00	93.97	8.22	92.27		
	C	Pre	55.00	107.50	89.88	14.15	91.88	3.30; 0.001	14.6 dB HL
		Activation	75.00	110.00	103.91	10.26	108.80		
		12 mo post activation	80.45	110.00	104.43	10.47	110.00		

Legend: Min, minimum; Max, maximum; M, mean; SD, standard deviation; Me, median.

**Table 2 pharmaceuticals-15-01176-t002:** Hearing preservation 12 months after CI activation according to treatment regime.

		No Measurable Hearing	Minimal	Partial	Complete
**ADVANCED BIONICS**	Intravenous group (IV)	1 (9.1)	1 (9.1)	7 (63.6)	2 (18.2)
	Oral and IV group	1 (7.7)	2 (15.4)	8 (61.5)	2 (15.4)
	Control group	7 (43.8)	2 (12.5)	5 (31.3)	2 (12.5)
**OTICON**	Intravenous group (IV)	17 (81.0)	0 (0.0)	1 (4.8)	3 (14.3)
	Oral and IV group	5 (35.7)	4 (28.6)	3 (21.4)	2 (14.3)
	Control group	9 (60.0)	1 (6.7)	5 (33.3)	0 (0.0)
**MED-EL**	Intravenous group (IV)	4 (18.2)	1 (4.5)	12 (54.5)	5 (22.8)
	Oral and IV group	1 (4.8)	2 (9.5)	11 (52.4)	7 (33.3)
	Control group	9 (64.3)	2 (14.3)	2 (14.3)	1 (7.1)

Data are given as the number of patients (numbers in parentheses show percentages in each row).

**Table 3 pharmaceuticals-15-01176-t003:** Characteristics of the patients in each of the groups.

Type of Implant System	Characteristic	Intravenous Group	Oral and Intravenous Group	Control Group
Advanced Bionics	No of patients	11 (7 M, 4 F)	13 (4 M, 9 F)	16 (8 M, 8 F)
	Age (years): range, Mean (SD)	34–71; 51.2 (10.7)	28–84; 61.4 (13.4)	20–76; 61.6 (13.9)
	Operated ear	7 R, 4 L	10 R, 3 L	6 R, 10 L
Oticon	No of patients	21 (12 M, 9 F)	14 (6 M, 8 F)	15 (7 M, 8 F)
	Age (years): range, Mean (SD)	32–73 61.2 (11.1)	43–86 64.4 (10.0)	53–86 67.7 (9.7)
	Operated ear	9 R, 12 L	8 R, 6 L	5 R, 10 L
Med-El	No of patients	22 (12 M, 10 F)	21 (10 M, 11 F)	14 (7 M, 7 F)
	Age (years): range, Mean (SD)	26–68 48.8 (14.9)	24–78 56.2 (15.8)	20–73 49.4 (16.0)
	Operated ear	13 R, 9 L	9 R, 12 L	7 R, 7 L

Legend: M, male, F, female; age is expressed as range, mean (M), and standard deviation (SD); R–right ear, L–left ear.

## Data Availability

Data is contained within the article.

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
