# Peer review of "The Clinical Effect of Steroids for Hearing Preservation in Cochlear Implantation: Conclusions Based on Three Cochlear Implant Systems and Two Administration Regimes"

_pharmaceuticals, 2022, doi:10.3390/ph15101176_

Round 1
Reviewer 1 Report
The authors observed the effect of perioperative administration of glucocorticoids on residual hearing protection in patients undergoing cochlear implant surgery through different administration methods, and also compared the conditions of three commonly used cochlear implants. The data of this study can provide some references for clinical work.
Comments:
1. There are many variables in subgroup 1 and subgroup 2, such as: hormone type, cumulative dose of hormone, total duration of administration time, administration method, time point of starting administration, etc.. Which variable plays the key role? Further analysis and discussion are recommended.
2. Table 2 shows that for the AB and Oticon subgroups, the hearing loss after combined oral corticosteroids (O+IV group) was greater than that in the intravenous group (IV group) . Please analyze possible reasons.
3. What are the main differences between each brand of cochlear implant electrodes? Does this difference lead to different hearing outcomes and lead to differences in electrode impedance in the no hormone group (control group)? Is it caused by the material of the electrodes or by differences in the shape of the electrodes or some mechanical properties of the electrodes? The necessary analysis is recommended.
Author Response
Dear Reviewer,
Thank you very much for the additional comments and suggestions. We have modified the manuscript according to the comments below and answers for all comments are in the attachment.
Corresponding author

Reviewer 2 Report
This manuscript carried out at a Polish tertiary otologic center prospectively assessed hearing outcomes in 147 cochlear implantees receiving different implant systems from three manufacturers and various types of concomitant glucocorticoid administration (i.v., combined oral and i.v., negative control). Hearing was analyzed preoperatively until a follow-up of 12 months was reached. Impedances and contralateral results were also determined. The partial and complete hearing preservation rates were higher in patients receiving steroids and one implant manufacturer appeared to perform worse than the other two.
In general, these findings are interesting in light of the recent developments regarding cochlear implantation and corticosteroids. Nevertheless, several points have to be clarified to improve the manuscript and allow the reader to fully grasp the details of every aspect described by the authors.
Major points:
- The study is presented as a prospective trial, but how do the authors explain the group differences in the Oticon group? 21 patients received intravenous drugs and only 14 received intravenous and oral drugs?
- Calculate if there are any differences between the groups (Table 1). E.g., the AB oral and intravenous group does not appear to be balanced in terms of sex: 13 (4 M, 9 F) and the age differences are already mentioned later in the manuscript, but not described here.
Minor points:
FORMATTING
- Affiliation(s) and correspondence email address have not been completed yet.
- Some parts of the text are incorrectly highlighted (green/yellow).
- Table 1: The abbreviation M stands for male and mean here. I would suggest to not abbreviation mean. Define that age is presented in years.
MATERIALS AND METHODS
- Lines 89-92: It probably should be “at sound frequencies”, not “and”?
- Line 93: The disease is called “diabetes mellitus” in English.
- Please define again how the pure tone averages were calculated: Air conduction threshold across all measured 11 frequencies (0.125-8 kHz)? It is mentioned in lines 124-126 and 151-153, but not in detail where the actual formula is discussed.
RESULTS
- Line 216: Typo: “Thisalso”
DISCUSSION
- Many typos in this section, please carefully read everything again.
- The first sentences of the discussion are basically identical with paragraphs above.
- Line 228: “wide using way of administration” – widely used?
- Line 236: “is also using in practice” – is also used?
- Line 239: “there is limits in” – there are limits in?
- Line 267: “the Oticon”
- Line 301: “is cause by” – is caused by?
- Line 305: “Med-el” – use the same spelling throughout the manuscript (Medel vs. Med-el).
- Lines 310/311: “supports the pharmacological the use of steroids”
- Add limitations of the study – e.g., the long-term loss of hearing after such EAS-type surgeries.
REFERENCES
- Many self-citations, some of which are not necessary (“Our methods followed those described by Skarzynska et al. 2018, Skarżyńska et al. 123 2021, and Skarżyńska et al. 2022 [6,15,16].”).
- References 10 and 12 are identical!
- Many references have been formatted incorrectly with some journal names abbreviated and others kept in the original format (“JCM”, “Hear Res” as well as “Hearing Research”, “Frontiers in Cellular Neuroscience”, altogether missing for reference 34). Please check the author guidelines again and format everything accordingly.
FIGURES
- Figure 3 includes a typo (“consequenty” instead of “consequently”) and is partially repetitive (“every single day”, “each day”)
- Figure 4 appears useless as it just includes one formula. In addition there are two different types of brackets around the % sign: [%}
- Figure 6: The abbreviations used for the CI manufacturers here are inconsistent with the rest of the manuscript.
Author Response

(The authors gave the same response as above.)

Round 2
Reviewer 1 Report
The authors have addressed most of the comments adequately.